# *Pride and Prejudice* in Brazil's Popular Culture: A Photonovel and a Soap Opera

Maria Clara Pivato Biajoli

Departamento de Letras, Federal Univesity of Alfenas, Alfenas 37130-000, Minas Gerais, Brazil; mariabiajoli@gmail.com

**Abstract:** Soap operas are an integral part of Brazilian popular culture and the daily lives of Brazil's people. In 2018, the biggest TV channel in the country, Globo, broadcast a six-month-long soap opera called 'Pride and Passion', centered on the story of the Benedito family and their five unmarried daughters, who live in the small village of 'Vale do Café' ('Coffee Valley') around the 1910s, surrounded by the rural aristocracy and its coffee plantations. The obvious inspiration is Austen's *Pride and Prejudice*, and its choice is an indication of Austen's growing popularity outside English-speaking countries. This adaptation, which incorporates characters from her other novels as well, is the quintessential amalgamation of cultures and media, combining a canonical author of the English language with a Brazilian TV genre commonly seen as 'lowbrow'. It was not, however, Austen's first incursion in Brazil's popular culture. During the 1960s and 1970s, photonovels were an extremely popular genre there, usually translated into Portuguese from Italian productions, as was the case of the 1965 *Pride and Prejudice* photonovel, sold as a literary supplement to a widely circulated women's magazine. This essay analyses both cases of different, although connected, adaptations of Austen, arguing that Austen's presence in Brazil was always mediated by the expectations and appropriation of new media, while showing that the dialogue with popular culture can only enhance our understanding of the 'global Austen' phenomenon and her appeal across time and cultures.

**Keywords:** Jane Austen; photonovel; *Pride and Prejudice*; popular culture; soap opera

## 1. Introduction

The translation of an author's work into several languages usually denotes its appreciation around the globe by different cultures. In the case of Jane Austen, translation of her novels, letters, and manuscripts are just the beginning. In fact, when many critics today speak of the rise of a "global Austen", they refer to the fact that she is not only being read, but being lived, experienced, and culturally recreated into new meanings everywhere. Moreover, if Austen as a popular culture phenomenon is not limited to English-speaking countries, then we must look at all these new and varied manifestations if we wish to understand this phenomenon's complex nuances. This essay aims to contribute with this effort by focusing on a small part of Austen's trajectory in Brazil.

British novels had a strong circulation in Brazil in the nineteenth century, particularly after the arrival of the Portuguese Royal family in 1808, fleeing Napoleon's invasion of Portugal, and the opening of Brazil's ports to trade with other nations (Schwarcz and Starling 2015, p. 174). As demonstrated by Silva and Vasconcelos:

> By the 1820s, shops that lent and sold books were opened and were soon followed by the foundation of membership and circulating libraries similar to those in Britain and France. These shops and libraries helped spread the circulation of the books sent by European book dealers to Brazil. ... These institutions started operating as important spaces of sociability, and thus encouraged the expansion of literate culture, while providing entertainment, information and instruction to their members and users. This process, which began in the Imperial

Court, soon extended to the provinces, with the diffusion of a multifaceted network consisting of public and membership libraries, as well as other popular institutions. (Silva and Vasconcelos 2014, p. 4)

The English novels available to buy or rent, however, were usually French translations or Portuguese translations from French editions, as indicated by studies conducted on the catalogues of bookshops and circulating libraries, private libraries' records in wills, and official records of customs houses of the nineteenth century (See, for instance, Schapochnik 2008). As many studies have shown, this process of translation from English to French (and equally from French to Portuguese) was never 'objective', i.e., the translators acted more as rewriters with a mission to adapt the works to the expectations of their countries (Trunel 2013), then as neutral (albeit impossible) bridges between two languages.

Austen's name, so far, has been located in two catalogues from nineteenth-century libraries in the city of Rio de Janeiro. First, in the 'Rio de Janeiro British Subscription Library', where records show that all her novels had been available to their members, at least since 1854, not only in English but, remarkably, in the first editions submitted to publishers by Austen herself, and, posthumously in the case of *Northanger Abbey* and *Persuasion*, by Cassandra and Henry Austen (Biajoli 2019). Secondly, Sandra Vasconcelos has identified the novel *A Família Elliot*—a Portuguese translation from the French *La Famille Elliot*, in its turn a translation of *Persuasion*—in the Portuguese circulating library 'Real Gabinete de Leitura Português' (Vasconcelos 2017, p. 125). However, in comparison to the number of editions available in many libraries from writers such as Charles Dickens, Walter Scott, and Anthony Trollope (Schapochnik 2014, pp. 106–8), we can assume that Austen was far from being popular or widely read in Brazil in this period. The place of honour goes to Maria Edgeworth, who made the top ten most-imported authors by booksellers in Rio (Abreu 2014). Significantly, the first Brazilian translations of Austen were published only in the 1940s, after the movie *Pride and Prejudice* (1940) and the success of other 'book-of-the-film' editions, such as *Gone with the Wind* and *Rebecca*, which encouraged a brief editorial line of collections of foreign classics or best-sellers translated from different languages (Hallewell 2017, p. 442). As was the case of the examples above, and of the two adaptations that will be analysed next, I argue in this essay that Austen's presence in Brazil has always been mediated by adaptations, meaning that the process through which she finally became renowned in this part of the tropics was composed of many chapters, each associated with a different cultural product that influenced her image here.

## 2. *Pride and Prejudice* in Pictures

Photonovels, or, in Portuguese, 'Fotonovelas', were an Italian creation from the period soon after the Second World War, when, as stated by Ermano Detti, Italy had 'a need to dream again' (Vecchia 2021, p. 9). Its origins can be equally traced to comic books and films because photonovels are a combination of both: stories told in pictures (only in the very beginning with drawings) of actors posing according to a script, which then would be arranged on the pages of a magazine in organized frames accompanied by short thoughts and dialogue as subtitles or in rudimentary speech bubbles. They also present small explanatory paragraphs that move the narrative from episode to episode (or microchapter to microchapter). They specialized, for the most part, in one genre only, sentimental stories, which identifies a third parent of this phenomenon: romantic novels—called 'romanzi rose', pink novels—and its very popular format, the serialized novels of the nineteenth century, the French *feuilletons* (Vecchia 2021, pp. 11–12).

The success of photonovel magazines was fast and vast, indicating a great potential to be exported to other countries. Soon, they arrived in Latin America, achieving huge popularity in Argentina, Mexico, and Brazil. Despite their popularity—or because of that— they were considered lowbrow, escapist, and inadequate reading material, particularly for young girls. The famous Argentinian cartoonist Quino registered this common opinion in his strip 'Mafalda', about a little girl with lots of grown-up questions. One day she asks her friend Susanita what she is reading, and when the answer is a photonovel, she interjects,

'Why, Susanita, don't fill your head with this nonsense. In the world many important things are going on, things that will change the destiny of mankind!' Susanita then cries: 'Don't remind me of it, you nag! Why do you think I read photonovels?' (Quino 2007, p. 301).

Juliana Ferreira de Melo conducted interviews with different readers of photonovels in Brazil from different groups (gender, race, level of literacy, and social class). She discovered that only young women would buy these magazines, while young men would borrow them from the women they knew (a sister, a housemaid). One woman, identified as Margareth in the research, remembered that she used to buy the magazines with the monthly allowance given by her parents, and then would lend them to her sisters, to a cousin, and to some friends from school (all girls), because their parents would not let them read photonovels (Melo 2017, p. 269). Two other participants, Ana Lúcia and Esther, said they only bought the magazine occasionally, their salaries as housemaids meant that they could not afford the price, despite these being considered cheap reading material (Melo 2017, p. 269). Moreover, their employers would forbid them to read the magazines because they saw them as 'indecent' and they would teach the girls 'nonsense' (Melo 2017, p. 269). Ana Lúcia and Esther, however, kept their reading habit, borrowing magazines from other friends, for example, or (secretly) reading the magazines their employers bought—indicating that, despite the alleged indecency and nonsense, the 'ladies of the house' also appreciated them (Melo 2017, p. 269). The incongruity between discourse and action in this case, between morally condemning photonovels and consuming them, is akin to how moralists of the eighteenth and nineteenth century obsessively censured female novel reading in England, while, according to Paula Backscheider, 'English prose fiction was', from the beginning, 'dominated by women' (Backscheider 2000, p. 4). Similarly, even with the many differences between the participants of Melo's research, they all appreciated photonovels and read them for many years, contradicting the assumptions (and prejudices) of the period that they were consumed only by women and by those who did not have high levels of formal education. The research also showed that statistic data of print runs are just the tip of the iceberg of the photonovels' reach, because there was a naturalized culture of borrowing that increased the number of readers of each issue. Everything considered, this was indeed a popular genre.

The first magazine to publish a photonovel in Brazil was *Grande Hotel*, a direct translation from one of the most famous Italian magazines (*Grand Hôtel*), in 1951. *Grande Hotel* followed the serialized format, releasing a chapter of a story in each issue, but soon another Brazilian magazine, *Capricho*, started publishing one complete story per issue. The success was immense and made its format the standard for all the other magazines of the same kind. *Capricho* was the second most-sold magazine in all genres in the country, the first place belonging to children's comic books (Disney's *Donald Duck*, *Mickey*, and *Uncle Scrooge*) (Habert 1974, p. 22). Soon, many publishers launched their own magazines to cater to the growing taste of their expected (if not exclusive) audience, women. The titles of these periodicals are telling enough of the genre of the stories and the target consumer: *Sentimental*, *Romântica* ('Romantic'), *Ternura* ('Tenderness'), *Cinderela*, *Meu Romance* ('My Romance'), *Sonho* ('Dream').

*Capricho*, however, innovated again when, in 1963, it released a quarterly supplementary photonovel to accompany the main magazine called *Supernovelas Capricho*. These were special photonovels because they were longer and had more pictures—therefore, more appeal. Soon, *Supernovelas* became a monthly publication as well. Issue number 8, published in 1965, announced a 'Fabulous All-New Photo Novel! 650 pictures!' The title was '*Pride and Prejudice*—The Story of an Unforgettable Love'[1]. The production was Italian, implicit in the names of the main actors, such as Patricia Del Frate as Lisa Bennet and Giacomo Rossi Stuart as Darcy[2]. In fact, it was rare that publishers in Brazil would produce their own photonovels[3], choosing, instead, the easiest and cheapest option of buying the copyrights, script, and pictures of the Italian magazines, translating the texts, and reorganizing the images according to their publishing formats and space constraints.[4] The technical information on the issue only identified the actors—there was no mention of

the director, scriptwriters, location, or year of the original Italian production. There was not any mention of Jane Austen either, or that the story was firstly a novel.

The plot follows the main narrative of *Pride and Prejudice*, however with many alterations and a large dose of melodrama: It opens with five unmarried Bennet sisters, daughters of a bourgeois country family, and three new arrivals in their neighbourhood: Mr Bingley, his sister Caroline Bingley, and his friend, a duke, Darcy, that are come to stay at the Netherfield Castle. The opening line locates the story 'At the beginning of the previous century, in a small province of England' (Orgulho e Preconceito 1965, p. 4). Although the information seems to refer to *Pride and Prejudice*'s original publication date, the costumes of the actors take the story into the Victorian period, much more similar to the aesthetic of the 1940 *Pride and Prejudice* movie. There is also an attempt to keep Austen's humour of the first famous sentence, now changed to 'the arrival of a newcomer was an exceptional event. If the newcomer was a single rich young man, things would escalate to fantastic proportions, especially in the homes where there were young ladies of age to marry' (Orgulho e Preconceito 1965, p. 4). Even with Lisa (Elizabeth) making fun of her mother's obsession to marry off her daughters, the story soon drops this tone and by the third page is already focusing on love and its obstacles.

Bingley and Jane are immediately and genuinely taken with each other, and Lisa also falls in love at first sight with Darcy. His feelings for her, on the other hand, will increase with time because of his inner struggles. His pride, however, hurts her, despite her acknowledgment that he is entitled to feeling above his company, 'being used to balls among the nobility' (Orgulho e Preconceito 1965, p. 9). Their class and social difference are 'an unsurmountable abyss' (Orgulho e Preconceito 1965, p. 9) between them, and Darcy's cold and distant treatment of her drives Lisa to tears on several occasions. She is torn as, despite showing him his incivility with witty words, Lisa is sad because it is a hollow victory: 'in truth, her heart would like to be humble and a slave of love' (Orgulho e Preconceito 1965, p. 10). When he proposes the first time, even though she loves him deeply, his pride and the lies told by Wickham will make Lisa refuse Darcy.

Caroline Bingley, angry because she has discovered that Darcy intends to marry his cousin, Ana du Bourgh, daughter of the Duchess du Bourgh, decides to thwart Jane and Bingley's relationship. In a dramatic scene, after convincing her brother to leave Netherfield, she visits Jane and lies to her, telling her that Bingley is going to marry Darcy's sister on the same day that Darcy marries Ana. Jane faints—her unconscious body is the focus of a large picture—and Lisa throws Caroline out of the house, saying: 'Go away! Can't you see that I hate you?' (Orgulho e Preconceito 1965, p. 54). The language of sensibility that, according to Janet Todd, was physically manifested in sentimental novels through tears, sighs, and faints (Todd 1986, p. 77), is also a visual language of photonovels, because photographs are the key element here. It is a language that Austen uses only critically, as with Marianne Dashwood's excesses, or mockingly, as in her Juvenilia. One could suspect that Lisa would like to faint alternately with Jane on the sofa after learning that Darcy is similarly going to get married soon, but her answer is also of feeling, with her strong, nearly violent, reaction against Caroline—we can almost hear the slamming of the door. Caroline then leaves without giving Jane the letter Bingley has written to say goodbye and reassure Jane of his return, but we can read his words: 'See you soon, my dear. Do not destroy my most sweet hopes' (Orgulho e Preconceito 1965, p. 54). Caroline rips the letter into pieces and throws them to the wind. The pieces of paper on the grass are the focus of the last picture in this page, with the caption 'His [Bingley's] hopes will never be answered' (Orgulho e Preconceito 1965, p. 54). The lovers would be kept separate because of the machinations of a spiteful young woman.

In the end, however, all will be fixed, as it should. Even the elopement of Wickham and Lydia is a happy one. Wickham says to Lisa that he 'courted Lydia from a whim, eloped with her for gain, married her for love' and admits, 'I was embarrassed by myself. Was it a miracle of love? I don't know' (Orgulho e Preconceito 1965, p. 80). The formula of sentimental novels, re-enacted in sentimental photonovels, does not allow for sad endings

for those undeserving of it. Thus Lydia, a naïve victim who does not resemble the 'untamed, unabashed, wild, noisy, and fearless' (Austen 2006b, p. 348). Lydia of the novel, is saved from a marriage to a bad man, and Wickham is the incarnation of the 'redemption through love' argument. He may not be able to say for sure, but readers will see it, indeed, as a miracle of love.

Soon after, Bingley and Darcy return. Jane and Lisa, who were out walking, see them arrive and are 'frozen, incapable of speech' (Orgulho e Preconceito 1965, p. 82). Bingley hesitates but then takes Jane into his arms exclaiming 'My Love!', to which she answers, 'You have returned, Charles!' (Orgulho e Preconceito 1965, p. 82) After Jane and Bingley's reunion, Darcy approaches Lisa; they talk about *her* misunderstanding of his character and about his generous actions towards Wickham to promote her sister's marriage, and then she apologizes. He does not. It is not necessary, as this hero has not been 'properly humbled' (Austen 2006b, p. 410) by Lisa, he is only a victim of her misconceptions and Wickham's libel. Darcy hesitantly proposes again, and Lisa answers: 'I have always loved you. I hated myself when I hated you. I suffered when I saw you suffer. All I wanted was to be with you' (Orgulho e Preconceito 1965, p. 82). Then the sky is shining, the rustle of the leaves 'sounds like a loving sigh', and they embrace. Darcy concludes with the wish 'to be with you, for all my life' (Orgulho e Preconceito 1965, p. 82). It was a 'tenderest and completest éclaircissement' (Austen 2006a, p. 229), to quote Austen's 'Plan of a Novel'—which could work just as well as a 'Plan of a Photonovel'.

From these examples, it is clear that this is an adaptation that incorporates very little of the novel besides the main plot. There is no irony, nor does the format of the photonovel allow for a critical narrator to show the inconsistencies of the characters. Their thoughts are revealed only to display their emotions—pain throughout the story, and happiness in the end. Because I have not been able to discover further information regarding the original Italian production, one question remains: did its writers see their adaptation as altering Austen to fit into the sentimental formula of photonovels, or did they believe her novel was already a 'romanzo rosa' and, therefore, needed only 'lops and crops' to fit the size of a magazine? The answer could tell us a lot about Austen's reception outside academia in the first half of the twentieth century. Given the changes affected in the 1940 *Pride and Prejudice* movie, which transformed the novel into a light comedy,[5] and the close ties of photonovels with films, it would not be a surprise if it were the latter case. Additionally, it is very interesting that the first Italian translation of *Pride and Prejudice* was published by the Mondadori house in 1932,[6] whose own photonovel magazine, *Bolero Film*, coined the term 'fotoromanzo' 15 years later (Vecchia 2021, p. 11). There seems to have been, at least in the activities of this specific editorial house, a proximity between publishing novels and publishing photonovels.

According to the research conducted by Angeluccia Habert in a period when photonovels were still strong in Brazil, the average print run of *Capricho Supernovelas* was around three-hundred and fifty thousand copies per issue (Habert 1974, p. 26). Given the popularity of these magazines and the borrowing system studied by Melo, we can assume that many people read this adaptation of Austen in the 1960s in Brazil without knowing that it was based on her work. Since photonovels were seen as lowbrow and disposable products, meant to be read and discarded after the next issue arrived at newsstands,[7] they were also meant to be forgotten—the recycling of the formulaic repetition of these melodramatic stories depended on it. At the same time, by this decade, only four novels by Austen had been translated in Brazil, the golden era of English translations commissioned by publishers long gone (Hallewell 2017, p. 508). Did readers of photonovels and of these four translations intersect? Could someone have recognized the *Supernovela* story and its author? So far, it has been impossible to tell. It seems, however, that Austen's first incursion in Brazil's popular culture was silent and anonymous, without her becoming more renowned because of it. It was a very different story when she was summoned back for a second run in 2018.

### 3. Pride with More Passion

As I stated earlier, Austen's presence in Brazil has been highly influenced by mediation: first, with the arrival of the French-Portuguese translation of *Persuasion*; second, with the short window of Brazilian translations of some of her novels after the 1940 movie; third, with the translation of an Italian photonovel adaptation of *Pride and Prejudice*. This is not different from the current 'Austenmania', which has also reached the tropics, although a little late. This specific story began in 2005. Before that, Austen was not very famous in Brazil, despite all previous films, and that was reflected in her material presence. It was difficult to find the DVDs of her adaptations, for example, while translations remained scarce and new editions rare. However, the 2005 *Pride and Prejudice* movie came and changed everything. It was an instant hit. Suddenly, Austen was popular among the female audience, who soon became her readers and her fans. The first groups of Brazilian fans organized themselves on social media, and in 2009, a small assembly decided to found the Jane Austen Society of Brazil—JASBRA. The success of the movie also had an effect on the availability of other Austen-related products. Within a relatively short period, there were several editions of her novels, different new translations, and, finally, DVDs of adaptations, recent and old, of all her completed novels.

It is important to note, then, that Brazil really began to read Austen after seeing the movie(s), and this new mediation will have marked her image. As stated by Katie Halsey, readers do not limit themselves to the written text to form their impressions: 'Nobody reads in a cultural vacuum, and reading can never be innocent of the influences of social, political, and economic structures, both those of the moment and the past' (Halsey 2013, p. 9). If Austen's popularity in Brazil today is a consequence of movie and TV adaptations, then this is how people will read her novels, based on what they watched. The adaptation informs the text. It is my belief, therefore, that this mediated image of Austen in general, and *Pride and Prejudice* in particular, led to her being considered as the basis for a soap opera.

Soap operas are a TV genre strongly connected with the serialized novels of the nineteenth century, particularly sentimental novels. In fact, TV soap operas are the latest—and most long-lasting—product of Brazil's taste for melodrama that evolved from the stories published in newspapers to radio soap operas, to photonovels, and finally to TV. The main producer and broadcaster of soap operas is the TV channel Globo. During its weekday schedule, there are three time slots reserved for new soap operas: 6 p.m., 7 p.m., and 9 p.m. The 6 p.m. soap opera is usually a comedy. It is lighter, funny, with some exaggerated comic characters, and is not too explicit with sex or violence. Commonly, these are period pieces and literary adaptations. For example, there has been a soap opera based on *The Taming of the Shrew* that achieved such great success that it has been rerun many times since in the mid-afternoon time slot reserved for that purpose.[8] The first thing we must notice, then, is that Austen was chosen for the 6 p.m. slot and not for the 9 p.m., for example, which is usually dedicated to darker or more serious topics—it shows how she is understood in Brazil.

The name of the soap opera was *Orgulho e Paixão* ('Pride and Passion'), and it aired from March to September 2018. The trailer released by Globo to publicize the new production offers a good summary: 'Coffee Valley . . . the twentieth century is just beginning. And if it depends on my matchmaking skills, it will begin well indeed for the Benedito sisters, the fairest of the town' (TV Globo 2018, 0:05–0:15). This was narrated by Ema Cavalcante, a secondary character inspired by Emma Woodhouse. She intends to help her neighbours and friends, the five Benedito sisters, to find husbands. Then she proceeds to present them: 'Jane is the most beautiful. Mariana, the most adventurous. Cecilia, an avid reader. Lydia, the high-spirited. And Elisabeta . . . well, Elisabeta is more contrary' (TV Globo 2018, 0:25–0:44). The family is based on the Bennets, from which we have Jane, Elizabeth, and Lydia. Mary Bennet is replaced by a character representing Marianne Dashwood, and Kitty Bennet is replaced by Cecilia, representing Catherine Morland. Each sister has a unique personality and prominent trait, an exclusive colour for their dresses, and a theme song, and each will live their own subplot (love) story—including Ema herself. The

main character, however, is Elisabeta. Through Ema's words, we learn that she is different. In the ensuing dialogue, they discuss marriage:

> [Elisabeta] 'I never said I didn't want to find a great love and get married. It is one of the things I want for my life, yes.'
>
> [Ema] 'Isn't it *the* thing?'
>
> [Elisabeta] 'It isn't *the* priority.' (TV Globo 2018, 0:45–0:53)

The idea of a young woman at the beginning of the twentieth century seeking a career instead of a husband may have been founded in the current perception of the original heroine of the novel. We cannot deny that Elizabeth is a young woman who defies, up until a point, the female roles and constraints of the period. Twice she gives up the chance to save her family—when she refuses Collins, and when she refuses Darcy. On both occasions, as Collins himself says, she could not know for sure that other proposals would come, and there is no way out of the entailment of Longbourn. Elizabeth risks the possible welfare of her family after her father's death for her individual happiness. We are spared the reflection of how selfish this is by her marriage to Darcy, and Jane's to Bingley, which ensure that the Bennet women will be taken care of. However, in contemporary times, after going through three or four waves of the feminist movement, this show of independent thought is never questioned nor condemned; on the contrary, it explains why Elisabeth is such a beloved character and Austen is seen by many today as a feminist writer (See, for instance, Auerbach 2004, pp. 7–8; Wells 2011, p. 150). Brazil's Elisabeta voices some questions historically raised in the 1910s, such as the right of women to work. Because this point has already been accepted in our society since then, her wish to pursue a career was not frowned upon. However, her wish to seek this dream instead of getting married, or even her defence of a woman's right to choose whether to be a mother, could have been more controversial, particularly in a country still too conservative[9], if this outspoken young lady were not safely ensconced in a soap opera, where, we all know, the main heroine always gets married in the end and has (or adopts) children, as was the case of Elisabeta and her four sisters. For the more progressive audience, on the other hand, her views were positive because they resonated with their own beliefs—in some cases, it might have been a confirmation of their own perception of Austen's character.

Returning to the trailer, we see Elisabeta riding in the hills and taking in the breathtaking view before her, as she holds a globe and murmurs to her horse: 'I'll see the whole world, Tornado' (TV Globo 2018, 1:10–1:14). *Orgulho e Paixão* embraces the outdoor images we have come to expect from Austen's adaptations since the 1995 *Pride and Prejudice*, in a direct dialogue with these previous productions. In particular, various scenes of Elizabeth 'on top of the world' from the 2005 movie are used to show how she craves freedom and new horizons. Her uniqueness is also visually marked in the first chapter when they all go to a costume ball given by Ema for all the single men and women of the neighbourhood. All women wear large gowns, reflecting the fashion of the previous century and (again) the Victorian era, but Elisabeta arrives dressed as a man. It is an extreme take on the famous petticoat with six inches deep in mud.

But then, Elisabeta meets a man called Darcy Williamson, a railroad builder and friend of the new owner of the local coffee plantation, Camilo Bittencourt (Charles Bingley). They argue from the start, but they both feel immediately attracted to one another. Elisabeta is very conflicted because she does not want to get married right away, she does not want to fall in love because it might be an obstacle to her goals. At the same time that she does not like Darcy very much because of first impressions, she daydreams about dancing with him, kissing him. She is in denial, for she is already in love with him, and he with her, as the audience already knows. Their shared problem, indicated in the title, is the conflict between their own pride *versus* the passion they feel for each other, and their first kiss, in chapter five, is a moment of temporary surrender that advances how their courtship will develop in fits and starts. If in the original novel it is necessary that both Darcy and Elizabeth grow to overcome their pride and prejudice so they can meet again in the end on equal

terms, the replacement of prejudice with passion in this adaptation is another key element that showcases, once more, an interpretation of Austen that focuses on the love plot and transforms complex characters into perfect (simplistic) ones. Therefore, their happily ever after is not prevented by their own faults but, firstly, by their mutual resistance, by many misunderstandings, and, occasionally, by violent schemes orchestrated by the villains. For instance, because they do not need to change, it is important that Darcy is never against Elisabeta's feminism and does not curb her dreams. By the end of the story, she has become a published writer (a nod to Austen, maybe) and has travelled the world with him—so, for a woman ahead of her time, only a progressive man (shown by his work with industries) would do.

Another key aspect of this soap opera deserves some attention. Why the change of the year of the story from something closer to the original (for example, the date of the publication of *Pride and Prejudice*) to a hundred years later? The answer might be in Brazil's history and our inability to deal with a violent past of slavery. In the 1810s, Brazil was still a colony of Portugal. Because it was already a large territory, the economic activities and social structure of the period were diverse; overall, however, one thing was true: slavery was everywhere (Klein 2018, p. 191). Rich people were either landowners or involved in different kinds of trade, but they were always white, and all of them were enslavers, meaning that they were either involved directly in kidnapping and transporting enslaved people, for large profits, or that they bought enslaved people and forced them to work in their farms, their shops, and their homes. Some even had 'slaves to rent', who could be hired to work for someone else for a specific task for a fee—paid, obviously, to the master. In this context, therefore, a man with Mr Darcy's status and fortune must be an enslaver, as well as Mr Bingley and the Bennets, even if they were not as rich as Darcy and their income came from different sources.

If the waves of the feminist movement changed how readers perceive Elizabeth Bennet, so did the civil rights movements worldwide make it unacceptable, even in a country still rife with racism, that a hero could be involved in this, much less the perfect Mr Darcy. The subject *per se* is not taboo. There have already been many soap operas that were set during the eighteenth and nineteenth centuries[10] that depicted that tragic reality, but the hero must never be an enslaver. Other secondary characters can be portrayed as 'ignorant', in other words, even though they accept slavery as a given in the beginning, they are good people at heart and they will learn and change their views, usually becoming abolitionists too. The villains, on the other hand, will assume the image of a cruel, bloodthirsty 'slave master'. When the writers changed the setting of *Orgulho e Paixão* to the beginning of the twentieth century, they avoided the question altogether—prejudice was removed even from the title. The result was a too white soap opera, though the opposite was to be expected if the screenwriters were to follow demographical statistics of the period. According to a national census conducted in 1872:

> the country had a population of nearly 10 million people, of whom about 8.5 million were free and 1.5 million remained slaves. Regarding the racial composition, 38 percent were white, approximately 20 percent black, more than 38 percent pardos (mixed race), and 3.9 percent indigenous. People of African descent (blacks and pardos together, including all social conditions–that is, free, freed and slave) comprised 58 percent of the total population, or approximately 5.7 million people. (Chalhoub 2018)

Where were these 5.7 million people? The abolition of slavery did not make them disappear, despite the predictions (and hopes) of some 'scientists' of the period (Klein 2018, p. 343). However, they did not make it to this soap opera either. This whiteness, the effacement of people of colour, seems to be associated with Austen and the Regency Era (Kumar 2021) and to be almost expected of adaptations, as the debates and lamentations regarding the colour-conscious casting of the Netflix TV series *Bridgerton* (2020–) might attest.

Therefore, Darcy could never be the man described by Mrs Reynolds as a kind master to tenants and servants if the only people working on his lands and houses were there as

forced labourers. On the other hand, in the 1910s, Brazil had already abolished slavery, and immigrants from Europe, mainly from Italy and Spain, were brought to work on the crops (Klein 2018, p. 232). Contrary to the disappearance of the black population, these immigrants were represented in the soap opera from the start (they were white), and Ema will end up marrying a poor Italian worker. The country was also changing economically, and the industry was rapidly developing. This context opened an opportunity for the adaptation. Instead of representing traditional England by being a wealthy member of the landed gentry, this Darcy represents modern times. He is an entrepreneur, an industrialist. He builds railroads, which were constructed to move the coffee harvests from the country to the ports of Rio and Santos, therefore connecting his business with Bingley's. Brazil's industrialization was paid for with coffee money, as well as the development of cities (Klein 2018, p. 327). The 1910s was the peak of Brazil's *belle époque*, the modernization of towns and urban life. A perfect scenario for the rise of snobs such as Caroline Bingley—in this soap opera, she is actually a widow called Susana, a reference to *Lady Susan*. However, history has also shown us that working in factories during the nineteenth century and the beginning of the twentieth was not easy. Unions, protests, and strikes were marks of exploratory, unhealthy, and dangerous working conditions (Klein 2018, p. 336). In the same way that Darcy could not be a slave master, he could not be a heartless factory owner either. So, in this story, he respects the workers, pays them well, and puts one of his men, who had had an accident, into his own car to drive him for hours to the city hospital to save his life.

Furthermore, the golden era of the coffee plantations in Brazil seems to offer another key point to adapting Austen nowadays: it is a period in the past easy to look at nostalgically, much like Austen's Regency. Before Brazil became a Republic, we were an Empire. During this period, coffee was the main export, which created a new generation of rich landowners who were given noble titles by the crown and became generally known as the 'Coffee Barons'—Brazilian Ema is the granddaughter of such a man. These families had their big houses in their estates, but they also built townhouses, the city of São Paulo was one favourite destination. This scenario has all the 'right' elements of Austen's time: the city and the country, the landed gentry and the charms of urban life, without the necessity of addressing the wound of slavery. Therefore, it was a perfect setting for a soap opera based on a nostalgic reading of the past. At the same time, because of the progress defended by Elisabeta regarding women's rights, and the economic progress associated with Darcy's railroad company, at least the depiction of the main couple is a nod to the future.

## 4. Mediated Austen

Who is Austen in Brazil today? First and foremost, she still is the 'book of the movie', or the book of the 2005 movie with those famous and dramatic scenes of a flexing hand, a rain-soaked proposal, and a reunion at dawn with the bonus of a slightly open white shirt—the equivalent of the lake scene for new generations of fans. Consequently, even if she has become important enough to merit several new translations, they are usually sold as romantic novels by a woman for women—the design of the covers show loving couples, period dresses, flowers, soft pastel colours, and words written in elaborate penmanship, and Janine Barchas has already shown us the impact of covers on the interpretation of novels.[11] After 2018, Austen was also the 'book of the soap opera', due to the power of a nationally broadcast soap opera of reaching millions of houses every day. All this means that, in the last decade or so, she has been read, watched, and talked about in an unprecedented way in this country, but through appropriations centred on the love plot and on the (sexy) appeal of the main couple, factors which are not concerned with Austen's contributions as a novelist to the genre.

Thus, as I argued throughout this essay, her presence in Brazil has always been mediated by other cultures and media, which transformed her work into raw clay to be modelled according to the expectations of her new destinations. The fact that not many people in the country can read her in the original language also means that even those who seek to read, do so through a translator's words, and not Austen's. So, is this

Austen in Brazil really Austen? Despite her new fame, despite new translations of never-before-published materials, such as her *Juvenilia* and her letters, hers is mostly a feminine popular culture phenomenon in Brazil, far from achieving the same recognition she enjoys in English-speaking countries as one of the most important novelists in the history of literature. Her mediated image is not of a writer to be studied and discussed, but of a writer of pleasure, of fantasy, of escapism, an opinion similar to what moralists said, and Austen mocked, about novels two centuries ago, about photonovels three decades ago, and continue saying about soap operas today. Even with their immense popularity, constant presence in most Brazilian homes, watched by men and women alike, and the stardom of their actors, soap operas are still considered lowbrow and 'housewife culture'. Because *Orgulho e Paixão* is the most recent local appropriation, some of that derogatory discourse inevitably ends up attaching itself to Austen's name, which limits the expansion of her reading public, but the slow rise of academic studies about her in Brazilian universities gives us hope. After all, if more and more students are seeking to write papers and dissertations on Austen in Brazil, it is *because*, not in spite of, their having met and fallen in love with her through popular culture. Because of it, then, we, scholars from non-English-speaking countries, can contribute to the understanding of one of the greatest writers of the English language.

**Funding:** This research received no external funding.

**Institutional Review Board Statement:** Not applicable.

**Informed Consent Statement:** Not applicable.

**Conflicts of Interest:** The author declares no conflict of interest.

## Notes

[1] The image of the cover is available in a post on the Jane Austen Society of Brazil's blog: https://janeaustenbrasil.com.br/2011/12/23/orgulho-e-preconceito-fotonovela/ (accessed on 15 March 2022).

[2] Another instance of the crossovers between films and photonovels, many actors and directors from the Italian movie industry also worked in photonovel productions.

[3] Only one magazine, *Sétimo Céu* ('Seventh Heaven'), among dozens in circulation in the decades of 1960 and 1970, would consistently publish photonovels produced in Brazil. Such as was the case in Italy, the producers would also cast famous actors, but from national soap operas instead.

[4] In Brazil's case, during most part of the period of the publication of photonovels, publishers also had to edit texts and themes according to the censorship of the military dictatorship in the country (1964–1985) that defined what was morally accepted for circulation in various media.

[5] See, for example, Emily Auerbach's analysis of how the movie was written and advertised as a romantic comedy for women about women hunting for husbands, with several changes that erased social/class conflicts—even Lady Catherine became a supporter of Darcy's effort to win Elizabeth's hand (Auerbach 2004, p. 279).

[6] The official website of the Jane Austen Society of Italy (JASIT) offers a careful list of Italian translations and editions of all Austen's novels. See the page for *Pride and Prejudice* here: https://www.jasit.it/edizioni-italiane/orgoglio-pregiudizio/ (accessed on 15 March 2022).

[7] My own copy, however, seems to counter this common opinion among critics: the owner—impossible to identify the name—signed and dated his/her copy. Is that an indication that he/she was a collector?

[8] With the launching of a teenager soap opera called Malhação (1995–2020), in total, Globo broadcast five different soap operas every day for 15 years.

[9] Brazil's conservatism in this matter reflects in the struggle to change its legislation regarding women's rights. Abortion is still criminalized, with a few exceptions, and it was only in the last decade that the crime of feminicide was officially recognized.

[10] Brazil abolished slavery only in 1888, the last country in the American continent to do so.

[11] To understand why some teenage girls thought Mr Darcy was a vampire, see (Barchas 2019, p. xi).

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
