# Peer review of "Pride and Prejudice in Brazil’s Popular Culture: A Photonovel and a Soap Opera"

_humanities, doi:10.3390/h11040075_

Round 1
Reviewer 1 Report
This is an interesting and engaging essay which demonstrates a good understanding of notions of cultural adaptation over time, and applies these thoughts well to the specific case of Jane Austen in Brazil. The commentary on the 1960s photoserials and the comparison with contemporary soap opera adaptations of Austen’s characters and themes is generally well considered and relevant.
There is a good opening overview of the interpenetration of British/English literature from the early 1800s onwards which demonstrates aspects of European influence too. I would suggest that the author may want to reduce the extent of the bulky quotation from Silva and Vasconcelos that open’s the essay given this is more about the influence of European and French literature than English/Austen. I’d suggest that the opening section might have something more direct about notions of global Austen/Austen in international culture that could be drawn from recent studies here before contextualising the European influence on Brazilian culture in the 1800s. this would make the piece firmly centred on Austen from the outset, whereas at present that connection in the introductory paragraph feels more tenuous than it truly is. The suggestion – end of p.2 – that Austen’s influence and presence has always been intermediated by translation or film/visual as well as literary texts could have greater expansion given its central relevance to the overall argument which follows.
I would also suggest that the author undertakes some subsectioning within the essay – otherwise it can feel a little jumpy i.e. the abrupt transition from the context of Austen circulation in the period 1800-1940 at the end of p.2 immediately into a description of photonovels (p.3): a subheading separating out the discussion here would be helpful navigation (similarly on p.7 where the essay moves to the more recent adaptation; same on p.12 where we turn to a conclusion with the statement “Who is Austen in Brazil today?”). On the photonovels and female readers issue, reference here could be more explicit to the ways in which the novel in the 19th century was viewed partly as a gendered form too – see works like Kate Flint’s The Woman Reader, for example.
There are some stylistic issues that need correction – tense and word choice in places (“readers don’t restraint themselves” should be restrain p.8 etc).
Overall, I think the essay could be much stronger if there was a more explicit engagement and focus on:
- adaptation as a form of creativity and appropriation: the conclusion around engagement with Austen for non-English speakers is relevant here, but there is no concentrated discussion of how this is then at the level of plot and character rather than style/writing
- timeshifts: the universality of the love story element etc is ripe for adaptations that are repeated over time, but there’s no sense in the essay at present about how those more frequent adaptations of Austen’s works (the multiple more recent films of Sense and Sensibility and/or Pride and Prejudice, or any of the number of BBC adaptations) have been factored into the cultural narrative in Brazil – perhaps they haven’t at all, but this needs to be acknowledged or explained at some level because this opens up questions of cultural selection as well as adaptation. If, for example, Brazilian viewers don’t watch the global/international film versions or the BBC worldwide productions of these texts which are more “authentic” to Austen but prefer the Brazilian soap opera versions then that is a very interesting cultural choice which would really enhance the discussion in the essay.
Author Response
From reviewer #1
- 1st quotation, from Silva and Vasconcelos, was reduced.
- A new opening paragraph was added to introduce the global Austen topic (1.24)
- The introduction of the main argument (Austen’s presence being always mediated) was further explained. (3.74)
- Headings were added on pages 3, 8 and 12.
- The female reading issue of photonovels was compared to the discourse about female reading of novels (4.117)
- It was highlighted that the new appropriations focus on the love plot and are not concerned with style/writing of the novels (13.461)
- Throughout the essay there were already references to indicate that the previous adaptations affected Brazil’s view of Austen and were incorporated in the soap opera. In particular, paragraph 10.338 was altered to make this point clearer.
- Finally, the essay was re-read by editor Sandie Byrne to review language usage.
The author would like to thank both reviewers for their carefully reading and thoughtful suggestions, which have made me see the need to address these questions with deeper analysis in the future. I am aware that the alterations presented only begin to cover your suggestions but given the time and space constraints I trust they are satisfactory for now.
Reviewer 2 Report
The manuscript displays original thought and is a welcome and refreshing addition to the world of Austen studies by identifying a gap in knowledge by focusing on Brazilian adaptations.
The strengths of the manuscript is particularly found within its interesting analysis of the soap opera, Orgulho e Paixao, and changing the year in which the story is set so that slavery will not be dealt with. However, the translation of this soap opera's name could be further explored. Changing the title from 'prejudice' to 'passion' is an interesting feature and one that could have been analysed further, particularly in relation to Deborah Kaplan's concept of 'harlequinisation' where Austen adaptations have been reduced to elements of romance and sexualisation to appeal to the mass female audience. This would also have strengthened the author's argument that photonovels were read by, and targeted to, young women. The author's argument that photonovels were considered low brow could again have been emphasised later in the article when writing about soap operas. Soap operas are considered part of lowbrow culture so, by Brazil's insistence on continuing to adapt Austen in lowbrow cultural media, how is Austen viewed by the masses? This concept is particularly interesting, and presents a juxtaposition argument, considering Austen wrote about the middle-class gentry and was read by the same audience in England. This has scope for further analysis.
On page 8 of the manuscript, the author states that the 2005 adaption of Pride and Prejudice changed how the Brazilian audience viewed Austen and she was not famous before this. However, this contradicts the very compelling and strong argument made that Austen translations were boasted after the 1940 adaption and young women reading the photonovels. I would suggest rethinking this claim regarding the 2005 adaption, or perhaps it would be more apt to suggest that this was the second wave of Austen hitting the masses or even the first time the Brazilian audience knew it was Austen (considering that Supernovas did not mention her by name).
On page 9 of the manuscript, the author states that during contemporary times, having gone through three of four feminist waves, that Elizabeth's independence is never question nor condemned. I would suggest that the author is more specific on this point particularly since the adaptation Bridget Jones's Diary clearly demonstrates that women are still condemned for being single and not married, and in many cultures and countries where Austen is read (e.g. South Asia) it is precisely this reason that the audience identify with Elizabeth - the condemnation that is felt by their communities for not getting married.
I believe some of the arguments in this manuscript need to be strengthened and additional points made - hopefully the suggestions that I have made will help the author with this. Moreover, a thorough edit is needed as there are minor grammatical errors throughout. However, I would like to say to the author that the original thought behind this manuscript displays insightful critical thinking and academic scholarship and I hope you will continue to expand and explore this avenue of research as it has great potential.
Author Response
From reviewer #2
- The change in the title from ‘prejudice’ to ‘passion’ was addressed (10.356-370)
- Soap operas still viewed as lowbrow was emphasized in relation to our understanding o Austen (13.476)
- The apparent contradiction on page 8 was corrected. In fact, the 1940 film boosted a short-lived interest in translations that did not affect Austen’s fame in Brazil. (8.258 and 3.71)
- The reception of Elisabeta’s feminism in Brazil’s context was specified (10.327)
- Finally, the essay was re-read by editor Sandie Byrne to review language usage.
The author would like to thank both reviewers for their carefully reading and thoughtful suggestions, which have made me see the need to address these questions with deeper analysis in the future. I am aware that the alterations presented only begin to cover your suggestions but given the time and space constraints I trust they are satisfactory for now.
Round 2
Reviewer 1 Report
The revisions undertaken to the first submission certainly handle my key concerns. There remain some issues about English language which should be picked up at the proofing/copy editing stage.